# Phase Separation and Pairing Fluctuations in Oxide Materials

**Andreas Bill** [1], **Vladimir Hizhnyakov** [2], **Reinhard K. Kremer** [3], **Götz Seibold** [4,*], **Aleksander Shelkan** [2] **and Alexei Sherman** [2]

1   Department of Physics & Astronomy, California State University Long Beach, Long Beach, CA 90840, USA; abill@csulb.edu
2   Institute of Physics, University of Tartu, 1 W. Ostwaldi Street, 50411 Tartu, Estonia; hizh@ut.ee (V.H.); aleksander.shelkan@ut.ee (A.S.); alekseis@ut.ee (A.S.)
3   MPI for Solid State Research Heisenbergstraße 1, 70569 Stuttgart, Germany; rekre@fkf.mpg.de
4   Institut für Physik, BTU Cottbus, P.O. Box 101344, 03013 Cottbus, Germany
*   Correspondence: seibold@b-tu.de; Tel.: +49-355-693006

**Abstract:** The microscopic mechanism of charge instabilities and the formation of inhomogeneous states in systems with strong electron correlations is investigated. We demonstrate that within a strong coupling expansion the single-band Hubbard model shows an instability towards phase separation and extend the approach also for an analysis of phase separation in the Hubbard-Kanamori hamiltonian as a prototypical multiband model. We study the pairing fluctuations on top of an inhomogeneous stripe state where superconducting correlations in the extended *s*-wave and *d*-wave channels correspond to (anti)bound states in the two-particle spectra. Whereas extended *s*-wave fluctuations are relevant on the scale of the local interaction parameter *U*, we find that *d*-wave fluctuations are pronounced in the energy range of the active subband which crosses the Fermi level. As a result, low energy spin and charge fluctuations can transfer the *d*-wave correlations from the bound states to the low energy quasiparticle bands. Our investigations therefore help to understand the coexistence of stripe correlations and *d*-wave superconductivity in cuprates.

**Keywords:** phase separation; cuprate superconductors; electronic correlations

## 1. Introduction

Already in their groundbreaking paper on 'Possible High $T_c$ Superconductivity in the Ba-La-Cu-O System' [1] Bednorz and Müller discussed the possibility of 'superconductivity of percolative nature' to explain their observation. It may be that chemical inhomogeneity was in their immediate line of sight but they also discussed granularity and 2D fluctuations associated with the superconducting wave function [1]. The discovery that high-temperature superconductivity results from hole doping of a 2D antiferromagnet stimulated Sigmund and his group at the University of Stuttgart in close collaboration with Hizhnyakov from the University of Tartu to study the problem of how doped charge carriers behave in a 2D magnetic insulating lattice. According to their initial ideas, doped charge carriers are stabilized in the dilute limit as localized magnetic polarons in a 2D fluctuating antiferromagnetic environment. On increasing doping concentration, such polarons condense to form larger clusters ('droplets') and above a critical concentration a percolating phase is formed, which then becomes superconducting [2–4]. This scenario got early support (see Figure 1) from experiments on lanthanum cuprate phases which showed that an

antiferromagnetic and a superconducting phase can exist simultaneously and their ratio can favorably be modified by thermal quenching experiments [5,6]. In particular, the comparison of field- and zero field cooled magnetization curves of $La_2CuO_{4+\delta}$ and $La_{2-x}Sr_xCuO_4$ demonstrated that it is rather the electronic component (i.e., magnetic polarons) which is affected by the thermal treatment.

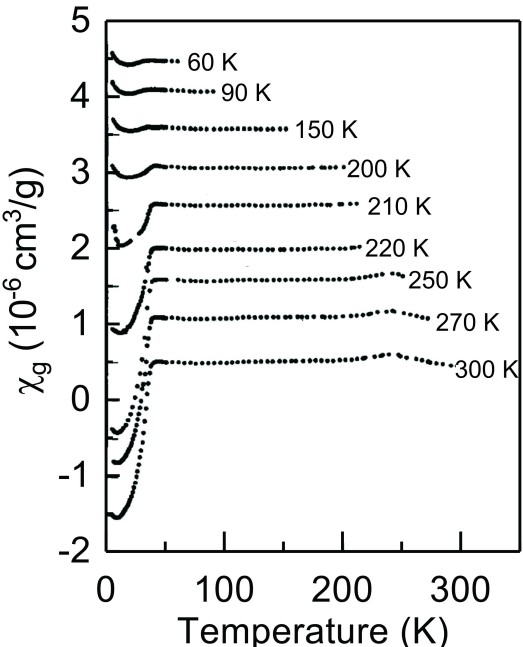

**Figure 1.** Gram-susceptibility of an 'as-prepared' $La_2CuO_{4+\delta}$ ($\delta \sim 0.01$) sample as a function of temperature. The sample was rapidly quenched from room temperature to the indicated temperatures and subsequently the magnetization ($B_{ext} \sim 9$ mT) was collected by slowly cooling the sample. Beginning from the lowest data set each curve was shifted upwards by a value of $5 \times 10^{-7}$ cm$^3$/g compared to the preceding one. (Adapted from Figure 1a, Ref. [5,6] by permission from Springer/Nature/Palgrave).

X-ray scattering experiments on analogously quenched $La_2CuO_{4+y}$ single crystals show that ordering of the oxygen interstitials in the layers of $La_2CuO_{4+y}$ is characterized by a fractal distribution of dopants up to a maximum limiting size of 400 μm which appears with the dopants enhancing superconductivity to high temperatures [7]. Evidence for charge segregation on a local scale came first from NMR [8] and NQR [9–11] investigations (cf. also Baranov and Badalyan as well as Hammel et al. in Refs. [12,13]). Independently, Emery and Kivelson emphasized that 'clumping' of the holes is an important feature of cuprate superconductors' [14]. Since this early experimental and theoretical evidence numerous experimental and theoretical accounts have appeared, discussing the importance of electronic inhomogeneity ('electronic phase separation') for high-$T_c$ superconductivity.

Instead of analyzing the formation of electronic inhomogeneities from the low doping side, an alternative theoretical approach is to investigate the phase separation instability of a correlated metal from the overdoped side, eventually supplemented with an electron-phonon interaction (see for example Refs. [15–20]). In this context, it was proposed [14,21,22] that the inclusion of long-range Coulomb interactions is a crucial ingredient since they suppress long-wavelength charge density fluctuations associated with phase separation favoring shorter-wavelength density fluctuations, giving rise either to dynamical slow density modes [14] or to incommensurate charge density waves [22].

Such incommensurate structures have been observed in $La_{1.48}Nd_{0.4}Sr_{0.12}CuO_4$ by Tranquada and collaborators who detected a splitting of both spin and charge order peaks by elastic neutron scattering experiments [23]. Their finding suggested that the doped holes arrange themselves in quasi-one-dimensional aggregates, 'stripes', which simultaneously constitute antiphase domain walls for the antiferromagnetic order. While the neutron scattering experiments only provide indirect evidence for charge ordering via the coupling to the lattice, bulk evidence for charge stripe order in the lanthanum cuprates has been found in $La_{1.875}Ba_{0.125}CuO_4$ and $La_{1.8-x}Eu_{0.2}Sr_xCuO_4$ by resonant X-ray scattering (RXS) experiments [24,25]. The rapid improvement and development of this technique has meanwhile led to the detection of charge order in a large variety of cuprate compounds, including YBCO, [26,27] Bi2212, [28] and Bi2011 [29]. Moreover, charge order was also measured in YBCO by high-energy X-ray diffraction [30] and quantum oscillations in both transport and thermodynamic experiments in magnetic fields [31–33] sufficient to suppress superconducting long-range order.

Whereas there appears to exist consensus on the formation of stripes in high-$T_c$ materials its relation to the mechanism of superconductivity is controversial. In fact, long before the discovery of high-$T_c$ Balseiro and Falicov [34] have shown that static charge-density waves (CDW) and superconductivity mutually suppress each other. Moreover, one-dimensional electronic correlations do not seem to be compatible with two-dimensional superconductivity in the high-$T_c$ compounds. On the other hand STM and ARPES experiments on LaBaCuO [35] suggest the existence of a $d$-wave gap below the stripe ordering temperature which is most pronounced for $\delta = 1/8$, when $T_c$ tends to zero. A subsequent study of the same compound presented evidence from the temperature dependence of the in-plane resistivity that this $d$-wave gap originates from superconducting fluctuations above a Kosterlitz-Thouless transition [36]. The authors conclude that the static stripe order is therefore fully compatible with two-dimensional superconducting fluctuations.

The essential role of electronic heterogeneities for superconductivity in hole-doped cuprates and the coexistence of multiple electronic components has been frequently pointed out by Alex Müller [37–39] in particular related to the formation and ordering of (bi)polarons [40,41]. For the particular case of stripes or CDW's there have been several attempts to link them to the pairing mechanism in high-$T_c$ superconductors. In a series of papers the Bianconi group has investigated pairing in a superlattice of quantum stripes where they found an amplification of superconductivity when the chemical potential is tuned towards a so-called shape resonance [42–47] and the multiband electronic structure can also induce an anomalous isotope effect [48]. In fact, formation of a CDW with the concomitant multiband structure can significantly enhance the intraband pairing scattering while suppressing the interband pairing [49,50]. However, inclusion of local Coulomb correlations has a strong impact on the renormalization of the electron-phonon vertex so that the interplay with CDW scattering can lead to both an enhancement or suppression of the pairing interaction [50]. Also the choice of the cutoff in the pairing interaction ('original' electrons vs. quasiparticles) plays a role in this regard.

Emery and coworkers have proposed a pairing mechanism [51] where holes on a charge stripe acquire a spin gap via pair hopping into the adjacent Mott insulating environment. Long-range superconducting phase coherence is then generated by Josephson coupling between the stripes. An alternative scenario has been put forward by Castellani et al. [22] It relies on the existence of a quantum critical point (QCP) near optimal doping. The QCP separates a homogeneous Fermi-liquid (in the overdoped regime) from a symmetry-broken ground state on the underdoped side of the phase diagram. The low doping phase was associated with incommensurate charge-density waves (ICDW). However, more exotic phases have also been proposed in this context. The singular fluctuations in the particle-hole channel generated in the vicinity of the QCP are reflected as divergent pairing correlations in the particle-particle channel. As has been shown in Ref. [52] an ICDW-QCP is compatible with a $d$-wave superconducting order parameter. More recently it was proposed [53,54] that superconductivity in the striped state occurs at a non-zero wave

vector ('pair density wave') which results in the suppression of the inter-layer Josephson coupling and thus a dimensional reduction in agreement with transport measurements on $La_{1.875}Ba_{0.125}CuO_4$. [36]

The aim of the present paper is twofold. First, we review in Section 2 the phase separation mechanism due to the formation and attraction of spin polarons. Section 3 is devoted to the problem how a phase separation instability in the Hubbard model can be realized without the additional involvement of phonons. Phonons (or other bosonic degrees of freedom) rather support the energy equilibration between the two phases which allows the phase separated state to be realized as a thermal state. Moreover, we show that the same mechanism can also be invoked to understand phase separation in multiband models including Hund exchange which is relevant for other oxide materials as for example manganites (cf. Ref. [55]). Furtheron, we show in Section 4 how isotropic superconducting correlations can be realized on top of an inhomogeneous electronic ground state. For this purpose we first review the pairing mechanism due to long-range optical phonon modes as proposed by Hizhnyakov and Sigmund [56–58]. We then exemplify the isotropy of superconducting correlations for a striped system where it turns out that for both $d$- and extended $s$-wave symmetry the corresponding vertex contribution has only a marginal orientational dependence.

## 2. Phase Separation in the Mean-Field Approximation

In the case of a homogeneous lattice, one of the sources of inhomogeneous charge distribution and lattice distortions (stripes) may be strong electron correlations. It was shown [59,60] that this phenomenon already takes place in the mean-field (Hartree-Fock) approximation of the three band Hubbard model.

In Refs. [59,60] we studied hole states in the antiferromagnetically (AF) ordered $CuO_2$ planes of cuprate perovskites with a self-consistent calculation of the Cu spin polarization. Both the Cu-O hybridization and the O-O transfer are taken into account. We used the following Hamiltonian for charge carriers (holes) in the $CuO_2$ plane, which follows from the original Hubbard Hamiltonian in the Hartree-Fock (HF) approximation:

$$H = \sum_\sigma H_{MF}^\sigma - U \sum_m \langle n_{m\uparrow}^d \rangle \langle n_{m\downarrow}^d \rangle, \tag{1}$$

where

$$H_{MF}^\sigma = \sum_n \left[ \epsilon_d + U \langle n_{n-\sigma}^d \rangle \right] n_{n\sigma}^d + \epsilon_p \sum_m n_{m\sigma}^p \tag{2}$$
$$+ T \sum_{nm} \left( d_{n\sigma}^+ p_{m\sigma} + \text{h.c.} \right) + t \sum_{mm'} \left( p_{m\sigma}^\dagger p_{m'\sigma} + \text{h.c} \right),$$

$d$ ($d^\dagger$) and $p$ ($p^\dagger$) are electronic annihilation (creation) operators on Cu and O orbitals, $U \approx 8$ eV, $T \approx 1$ eV, $t \approx 0.3$ eV, $\epsilon = \epsilon_p - \epsilon_d \approx 3$ eV. In the AF ordered $CuO_2$ plane the elementary cell is doubled (the magnetic unit cell contains two $CuO_2$ units ). The copper on-site energies are given by $\epsilon_{1\sigma} = \epsilon_d + U \langle n_{1-\sigma}^d \rangle$, $\epsilon_{2\sigma} = \epsilon_d + U \langle n_{2-\sigma}^d \rangle$.

In what follows we are interested in the behavior of a large-size wave packet of extra holes added to the AF ground state. The Hamiltonian (1) does not take into account the Coulomb repulsion of these holes, assuming that it is compensated by attraction with sufficiently mobile doping ions.

In the AF-ordered state $Cu_2O_4-$elementary cells form a simple square lattice with the lattice constant $a' = a\sqrt{2}$ and with main directions along $x' = (x+y)/\sqrt{2}$ and $y' = (x-y)/\sqrt{2}$. Therefore, it is convenient to use the site vectors $\vec{m}' = (m_{x'}, m_{y'})$ which count the elementary cells in the $x'$ and $y'$ directions ($m_{x'}, m_{y'} = 0, \pm 1, \pm 2, ...$; $\vec{m}'$ corresponds to the cell with coordinates $x' = a'm_{x'}, y' = a'm_{y'}$). Within this choice the second hole band (empty in the undoped case) has 4 minima at the points $(\pm \pi/a', 0)$

and $(0, \pm\pi/a')$ in the Brillouin zone. The wave functions of the minima contain only negligibly small $(< 10^{-6}/\sqrt{N})$ amplitudes of the first states $|d_1\rangle_{m'}$, corresponding to the Cu with the opposite spin; neglecting these contributions, the wave function can be presented in the form [for the minimum at $\vec{k}' = (\pi/a', 0)$]:

$$|\psi_{min}\rangle = \frac{1}{\sqrt{N}} \sum_{\vec{m}'} |\psi\rangle_{\vec{m}'}, \tag{3}$$

where $N$ is the number of elementary cells,

$$|\psi\rangle_{\vec{m}'} = (-1)^{m_{x'}} (\sin\alpha |d_2\rangle_{\vec{m}'} + \cos\alpha |P_1\rangle_{\vec{m}'}), \tag{4}$$

$$|P_1\rangle_{\vec{m}'} = \frac{1}{2}(|p_1\rangle_{\vec{m}'} - |p_2\rangle_{\vec{m}'} + i|p_3\rangle_{\vec{m}'} - i|p_4\rangle_{\vec{m}'}), \tag{5}$$

and $\sin\alpha \approx 0.39$ (for $U = 8T$, $t = 0.3T$, $\epsilon = 3T$); $|p_n\rangle_{\vec{m}'}$ denote the states of the 4 oxygens surrounding the second Cu ion in the $\vec{m}'$-th elementary cell, counted counterclockwise starting from the right position.

We construct the wave packet from the states close to $(\pi/a', 0)$, the minimum of the hole band. This wave-packet can be presented in the form

$$|\psi_L\rangle = \sum_{\vec{m}'} c_{\vec{m}'} a^+_{\vec{m}'} |0\rangle,$$

where $|0\rangle$ is the state with a filled lower Hubbard band, $a^+_{\vec{m}'}$ is a creation operator of the hole state $|\psi\rangle_{\vec{m}'}$, $c_{\vec{m}'}$ is the corresponding probability amplitude. We choose $c_{\vec{m}'}$ in the exponential form:

$$c_{\vec{m}'} = A_L \exp[-2\left(|m_{x'}| + |m_{y'}|\right) a'/L + i\pi m_{x'}], \tag{6}$$

where $A_L = tanh(2a'/L)$; the oscillating multiplier $exp(i\pi m_{x'} a'/L)$ accounts for the wave vector $\vec{k}' = (\pi/a', 0)$, of the $(\pi/2a, \pi/2a)$ minimum of the hole-band. This shape of the wave-packet is close to that of the soliton-type $(\sim sech(x/L))$ packet of the minimal energy for the given size $L = (\int |\psi|^4 dx)^{-1/2}$.

The expectation values of polarization are obtained from the self-consistent equations

$$\langle n^d_\sigma \rangle = \sum_k | \phi_{ik} |^2, \tag{7}$$

where $\phi_{ik}$ is the eigenvector of the Hamiltonian matrix, corresponding to the eigenvalues $E_k$. The second band (empty in the undoped case) has 4 minima at the $(\pm\pi/2a, \pm\pi/2a)$ points in the Brillouin zone.

Our solutions of the self-consistent Equation (7) show that for a single hole the lowest energy solution corresponds to a spin-polaron of small size [61,62]. The free hole state is about 0.15 eV higher in energy. We also found that in order to obtain such spin-polaron state from the state of a free hole it is necessary to overcome an energy barrier of about 0.05 eV before the formation of the polaron can occur [62]. However, at finite doping the spin-polaron states become less favorable and at a critical concentration $c \sim 0.5$ they turn out to be metastable.

We also have observed that already at small hole concentrations their spatial distribution changes from homogeneous to a domain type. Such behavior of holes is expected from general considerations. Indeed, in a two-dimensional lattice the self-energy of a large size $(\sim L)$ hole wave packet caused by the interaction with the surrounding Cu spins is $\propto |\psi|^4$. It depends on $L$ as $(-L^{-2})$, i.e., in the same way as its kinetic energy, but with different sign [63]. Therefore, in case of a high effective hole mass the attractive self-action dominates, thus leading to the formation of domains. According to our calculations the local hole concentration in domains is $\sim$0.5–0.6. This result is demonstrated in Figure 2, where, for a system

containing hole-enriched stripes, the dependence of the total energy on the concentration of holes in the stripes is shown for the case of a total hole concentration $c = 0.05$. The free energy of the domain only weakly depends on its shape. Consequently, the formation of stripe domains already takes place in the HF approximation. The optimum hole concentration in domains obtained via this approximation is close to the observed value of $c = 0.5$.

Similar results on phase separation were obtained using slave-boson, slave-fermion and large-N expansions [16,64–67]. Within the tJ-model it has been shown [68] that phase separation supersedes superconducting instabilities for large enough exchange coupling. Mechanisms of phase separation in solids different from cuprates (e.g., manganites) were considered in Refs. [55,69–71].

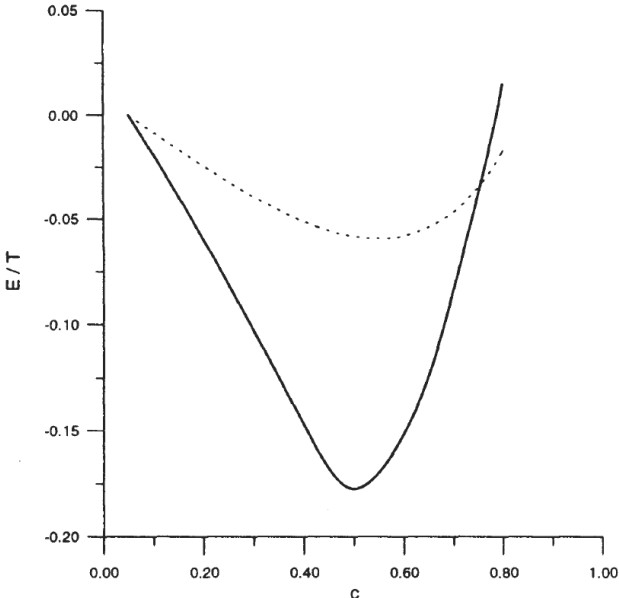

**Figure 2.** Full energy of the crystal $E$ vs hole concentration $c$ in the hole-rich (stripe) region. Initial (mean) hole concentration is 0.05. The dotted line corresponds to the rigid AF lattice.

## 3. Phase Separation and Fluctuations

In systems with strong electron correlations, phase separation takes place with the inclusion of charge and spin fluctuations. This result was recently demonstrated in the one-band repulsive Hubbard model on a two-dimensional square lattice [72]. It was shown that at low temperatures, regions of negative electron compressibility (NEC), $\kappa = x^{-2}(\mathrm{d}x/\mathrm{d}\mu) < 0$, arise near certain values of the chemical potential $\mu$. Here $x$ is the electron concentration. The source of this unusual behavior of $\kappa$ is the crossing of the energy levels in the Hubbard atom at these $\mu$. A power series expansion around the atomic limit is the natural investigation method in the case of strong correlations. This approach is called the strong coupling diagram technique (SCDT) [73–77]. The convergence of the series expansion is confirmed by the summation of infinite sequences of diagrams. Its validity follows from the successful comparison of its results with data from Monte Carlo simulations, exact diagonalization and experiments with ultracold atoms in optical lattices [76,77].

The Hamiltonian of the Hubbard atom reads

$$H_l = \sum_{\sigma}[(U/2)n_{l\sigma}n_{l,-\sigma} - \mu n_{l\sigma}], \tag{8}$$

where $U$ is the on-site repulsion, $n_{l\sigma} = a_{l\sigma}^{\dagger} a_{l\sigma}$ is the occupation-number operator on the lattice site $\mathbf{l}$ with the spin projection $\sigma = \pm 1$, $a_{l\sigma}^{\dagger}$ and $a_{l\sigma}$ are electron creation and annihilation operators. As seen from Equation (8), the Hamiltonian has four eigenvectors $|\lambda\rangle$: the empty state $|0\rangle$ with the eigenenergy $E_0 = 0$, two singly occupied degenerate states $|\sigma\rangle$ with the energy $E_1 = -\mu$, and the doubly occupied state $|2\rangle$ with the energy $E_2 = U - 2\mu$. As follows from the energy expressions, with the change of $\mu$, these states become alternately the ground states of the atom: for $\mu < 0$ it is $|0\rangle$, for $0 < \mu < U$ the degenerate singly occupied states are the lowest ones, and for $U < \mu$ the ground state is $|2\rangle$.

The terms of the SCDT series expansion for Green's functions are products of hopping constants and on-site cumulants [78] of electron creation and annihilation operators. In particular, the first-order cumulant $C^{(1)}(\tau) = \langle \mathcal{T} a_{l\sigma}^{\dagger} a_{l\sigma}(\tau) \rangle_0$, i.e., the first term of the expansion for the one-particle Green's function, after Fourier transformation reads

$$C^{(1)}(j) = \frac{1}{Z} \sum_{\lambda\lambda'} \frac{e^{-\beta E_\lambda} + e^{-\beta E_{\lambda'}}}{i\omega_j + E_\lambda - E_{\lambda'}} \langle \lambda | a_{l\sigma} | \lambda' \rangle \langle \lambda' | a_{l\sigma}^{\dagger} | \lambda \rangle, \qquad (9)$$

where $\mathcal{T}$ is the time ordering operator, the subscript 0 of the averaging brackets indicates that the averaging and time dependence are determined by Hamiltonian (8), the partition function $Z = \sum_\lambda \exp(-\beta E_\lambda)$ with $\beta = 1/T$ the inverse temperature, and $j$ is the integer defining the Matsubara frequency $\omega_j = (2j-1)\pi T$. At low temperatures, due to the Boltzmann factors $e^{-\beta E_\lambda}$ in Equation (9), the cumulant changes drastically as $\mu$ goes from a region with one of the mentioned ground states to another one. Namely, for $\mu \ll -T$, $C^{(1)}(j) \approx 1/(i\omega_j + \mu)$, while for $T \ll \mu$ and $T \ll U - \mu$, $C^{(1)}(j) \approx (1/2)[1/(i\omega_j + \mu) + 1/(i\omega_j + \mu - U)]$. In the third region, $T \ll \mu - U$, $C^{(1)}(j) \approx 1/(i\omega_j + \mu - U)$. Similar sharp changes occur in other cumulants. Since they enter into irreducible diagrams composing the irreducible part $K(\mathbf{k}, j)$, which defines the one-particle Green's function, [76]

$$G(\mathbf{k}, j) = \left\{ [K(\mathbf{k}, j)]^{-1} - t_{\mathbf{k}} \right\}^{-1}, \qquad (10)$$

sharp changes occur in spectral functions, densities of states, and band dispersions. Here $\mathbf{k}$ is the wave vector and $t_{\mathbf{k}}$ the Fourier transform of hopping constants. The drastic variation of electron bands near $\mu \approx 0$ and $\mu \approx U$ can be characterized as their pronounced non-rigidity—a strong dependence of the electron dispersion on the chemical potential/electron concentration. This non-rigidity is the origin of the NEC observed near these values of the chemical potential.

Figure 3 exhibits a cartoon image of one of the NEC regions. In the one-band Hubbard model, the topmost point of this dependence may be close to $x = 1$ at low temperatures [72]. Let us suppose that the crystal is divided into two parts with the electron concentration $x_1$ and chemical potential $\mu_1$ in one of them, and $x_2 < x_1$ and $\mu_2 > \mu_1$ in another. Representative points of these two parts are shown in Figure 3. Both parts are considered to be macroscopic crystals. Hence the dependence $x(\mu)$ is described by the curve in this figure. Let us suppose that we transfer an electron from part 2, with a smaller concentration, to part 1, with a larger concentration. Therefore, the concentration difference between the two parts is further increased. Such a transfer is energetically favorable, since $\mu_2 > \mu_1$. Therefore, if there is a subsystem in contact with the electron subsystem, which can absorb the energy $\mu_2 - \mu_1$, such an electron separation will proceed spontaneously until the concentration and chemical potential in part 1 reach the topmost point in the curve in Figure 3, while part 2 attains the lowermost point. The character of the curve prohibits further separation.

In crystals, such an energy-absorbing subsystem is provided by phonons. Let us consider the simplest model for vibrations – local lattice distortions $q_l$ linearly interacting with the electron density. This interaction is described by the term $H_i = -v \sum_{l\sigma} q_l n_{l\sigma}$. We use the adiabatic approximation. The two

parts of the crystal are large enough, which allows us to suppose that at the bottom of the adiabatic potential all local distortions are equal, $q_1 = q$. In this case, $vq$ in $H_i$ becomes a correction to the electron chemical potential, and the adiabatic potential is easily calculated. Its minimization gives $q = -vx/u$ with $u$ the elastic stiffness constant. Thus, the distortion in the electron-rich part is larger than in the electron-poor one. We arrive at a state in which both electronic components and distortions are inhomogeneous. If there is a predominant interaction of electrons with distortions of special symmetry, especially connected with a softening phonon mode, one can expect the formation of stripes.

Hence in our approach with a fixed chemical potential, the assistance of phonons is needed for phase separation. Ground states featuring phase separation were found in many works using different optimization procedures (see, e.g., Refs. [79–81]). These procedures do not fix the energy of a solution, but rather minimize it. Therefore, in such works, phase-separated ground states are obtained in purely electronic systems, without the involvement of phonons.

The charge instability in the form of the NEC is observed in other models of strongly correlated systems as well. In particular, we found such instability in the Hubbard-Kanamori (HK) model described by the Hamiltonian [82,83]

$$
\begin{aligned}
H \;=\; & -t \sum_{\langle \mathbf{ll'}\rangle i\sigma} a^{\dagger}_{\mathbf{l'}i\sigma} a_{\mathbf{l}i\sigma} + \sum_{\mathbf{l}i\sigma} \Big[ -\mu n_{\mathbf{l}i\sigma} \\
& + \frac{U}{2} n_{\mathbf{l}i\sigma} n_{\mathbf{l}i,-\sigma} + \frac{U-2J}{2} n_{\mathbf{l}i\sigma} n_{\mathbf{l},-i,-\sigma} \\
& + \frac{U-3J}{2} n_{\mathbf{l}i\sigma} n_{\mathbf{l},-i,\sigma} \\
& + \frac{J}{2} \big( a^{\dagger}_{\mathbf{l}i\sigma} a^{\dagger}_{\mathbf{l}i,-\sigma} a_{\mathbf{l},-i,-\sigma} a_{\mathbf{l},-i,\sigma} \\
& - a^{\dagger}_{\mathbf{l}i\sigma} a_{\mathbf{l}i,-\sigma} a^{\dagger}_{\mathbf{l},-i,-\sigma} a_{\mathbf{l},-i,\sigma} \big) \Big],
\end{aligned}
\tag{11}
$$

where the subscript $i = \pm 1$ labels two degenerate site orbitals and $J$ is the Hund coupling. Hamiltonians of this type are used for the description of transition metal oxides, iron pnictides, and chalcogenides. In contrast to the simpler one-band Hubbard model with only two NEC regions, in the HK model with two orbitals, there are four such regions. Two of them at $\mu = 0$ and at $\mu = U - 3J = 1.5t$ are seen in Figure 4. This dependence $x(\mu)$ was also obtained using the SCDT. The mechanism of the appearance of these NEC regions is the same, as in the one-band model, i.e., the level crossing in the on-site terms of the Hamiltonian (11) leads to sharp changes of electron bands (for more details see Ref. [84]). Indeed, phase separation is inherent in crystals, for the description of which the HK Hamiltonian is applied [85–87].

Hence the appearance of NEC regions corresponds to the common mechanism of phase separation in crystals with strong electron correlations.

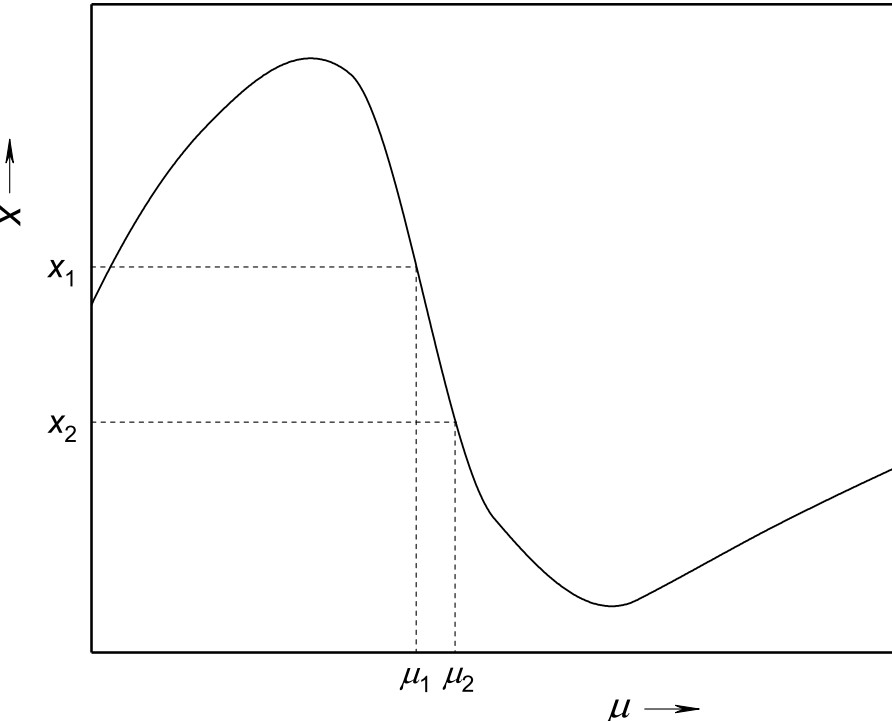

**Figure 3.** The dependence $x(\mu)$ near one of the NEC regions.

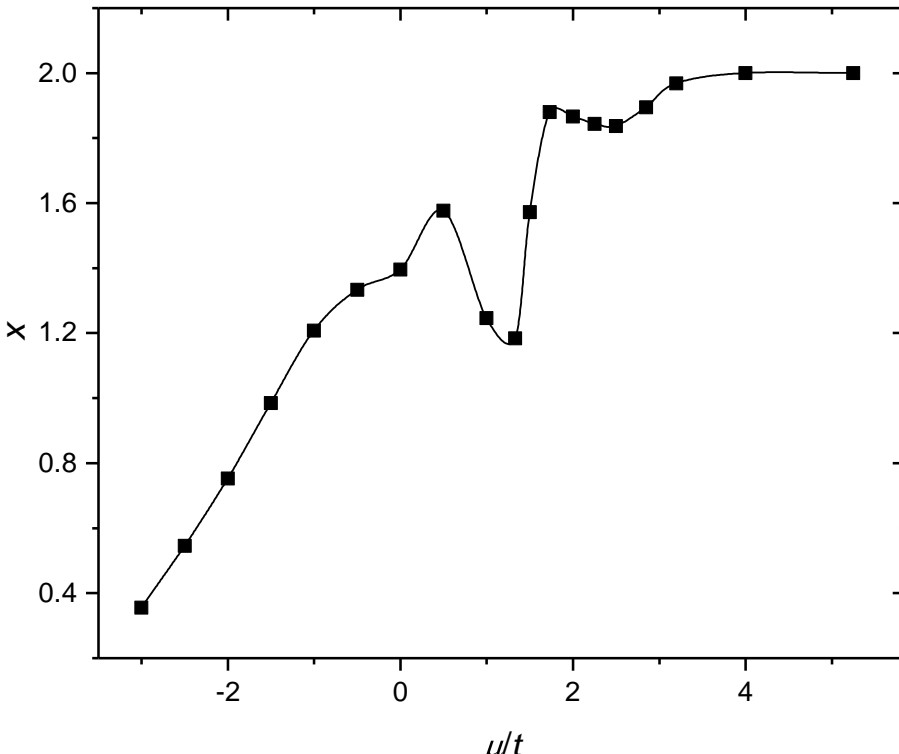

**Figure 4.** The dependence $x(\mu)$ in the Hubbard-Kanamori model with two orbitals. $U = 6t$, $J = 1.5t$, and $T = 0.13t$.

### 4. Pair Correlations In Cuprates

The presence of electronic inhomogeneities resulting from competing interactions at the microscopic level not only affects the normal state properties of strongly correlated materials. They also play a major role for the existence of the superconducting state. There have been many attempts at determining the pairing mechanism but no consensus has been reached in the community. As stated in the introduction, Sigmund and Hizhnyakov pointed out early on the importance of taking into account electronic inhomogeneities in the description of the superconducting state [2–4]. However, contrary to many they did not consider strong correlations or electronic inhomogeneities as the driving mechanism of superconductivity *per se*, but rather as modifying the conventional picture of superconducting pairing in ways that make the superconducting state of these materials unique.

At the time when Alex Müller visited Sigmund and Hizhnyakov in Stuttgart, he was very interested in the pairing mechanism proposed by the group and resulting from the transitive electron-phonon interaction in one-dimensional, percolative, stripe-like inhomogeneities [2–4]. With time, Alex moved his research in the direction of more local pair correlations, favoring the formation of bipolarons. He also viewed the symmetry of the order parameter as resulting from the coexistence of two condensates with *s*- and *d*-wave symmetry [88]. On the other hand, the authors developed a theory that involves electronic inhomogeneities and long range interactions in the formation of Cooper pairs [56–58]. As explained below, that model leads to a single condensate with anisotropic *s*-wave symmetry that changes into *d*-wave symmetry when accounting for the local Coulomb repulsion [56–58,89].

In the following we consider two aspects of superconducting pairing in a strongly correlated system. In the first, electronic inhomogeneities cause the electron-phonon coupling to have an essential long-range component that sustains a superconducting state while the balancing between the resulting effective attractive interaction with the local Coulomb repulsion determines the symmetry of the order parameter. In the second, the superconducting pair fluctuations are considered on top of a symmetry-broken ground state of stripes. The phase separated configuration and underlying interactions lead to pair correlations that appear very isotropic despite the anisotropic inhomogeneous electronic state. Common to both aspects of pair fluctuations is that they occur at low momentum transfer **q** but result from local electronic correlations which determine the symmetry of the superconducting order parameter.

### 4.1. Pairing from Long Range Electron-Phonon Interaction

Electronic inhomogeneities are the expression of static or dynamic phase separation into hole rich and hole poor regions. An essential aspect of these considerations is the time scale associated with different components of the phase separated state. It is less the absolute time scale that matters: the electronic inhomogeneities need not be static to result in the pairing mechanism discussed here. They may fluctuate but with a time scale much larger than the motion of quasiparticles within the metallic component of the inhomogeneous state. Furthermore, the inhomogeneities are on a microscopic scale and only in specific situations become more macroscopic as for example the stripe phase at $x = 1/8$ in LBCO.

The particular model used to describes the inhomogeneous electronic structure is not crucial for the considerations of this subsection. The important aspect is the reduction of screening effects. This reduction is reinforced by the layered nature and anisotropic transport properties of high temperature superconductors. As a result, there is poor screening along the *c*-axis, and the screening of the Coulomb interaction has an important dynamic contribution. Unlike conventional three-dimensional metals, low energy electronic collective modes appear in correlated layered materials [90–93]. These modes are acoustic ($\omega \sim q$), although they may display a small gap at $q \to 0$ that depends on the interlayer hopping parameter. The gap, however, does not affect the physics of low energy collective modes in an essential way [91–93]. Calculations demonstrate that low-energy plasmons contribute constructively to

the pairing mechanism in a variety of novel superconductors [91–93]. Only very recently have low-energy plasmons been observed experimentally using resonant inelastic X-ray scattering (RIXS) in electron doped cuprates [94,95].

The unusual plasmon spectrum resulting from electronic inhomogeneities, the layered structure and anistropic transport, have another implication that is essential for understanding the pairing mechanism in high temperature superconductors. The attractive electron-ion interaction screening is reduced in the hole poor region as compared to the hole rich phase. Hence, one expects two components to the electron phonon interaction, $H_{eph} = H_{eph,S} + H_{eph,L}$. The usual short-range, screened interaction $H_{eph,S}$ is dominant in most metals. The long-range, weakly screened interaction $H_{eph,L}$ is reminiscent of what occurs in a polar crystal, though high-$T_c$ materials are rather in an intermediate case between these two extremes. The long-range interaction $H_{eph,L}$ is unique to systems such as high-temperature superconductors since it results from the presence of strong-correlation induced dynamic inhomogeneities of the electronic ground state.

Breathing and buckling modes, which are the $A_{1g}$ in-plane and the $B_{1g}$, $B_{2g}$ out-of-plane motion of oxygen atoms in the $CuO_2$ planes, respectively, were shown to contribute most to the phonon-mediated pairing interaction [96,97]. It was argued that correlations suppress charge fluctuations, which leads to a reduction of the electron-phonon interaction [98,99]. This result was obtained for a local interaction. Calculations using the time-dependent Gutzwiller approximation have shown that correlations can enhance the transitive coupling to these optical phonon modes at small momenta **q** [100,101]. Nevertheless, the amplitude of the coupling to the $A_{1g}$ mode is too small in absolute value to lead to high critical temperatures. These considerations were made without taking into account the fact that the screening along the *c*-axis is reduced and allows for a three-dimensional coupling of charge carriers to long wave length optical phonons.

Sigmund and Hizhnyakov considered the pairing interaction resulting from long wave length optical phonons, noticing three essential features. First, the long range interaction implies an averaging over the smaller scale of electronic inhomogeneities and the interlayer distance, rendering the superconducting state truly three-dimensional. Second, the anisotropy of the superconducting order parameter does not result from the anisotropy of the pairing interaction which mixes states with close momenta **k**, but rather from the band structure and in particular the anisotropic density of states at the Fermi surface [56–58]. Third, the pairing interaction determines the stability of the superconducting state and magnitude of the superconducting order parameter. The symmetry of the latter is, however, not solely determined by the pairing interaction. Account of the local, Hubbard like, Coulomb repulsion and the relative magnitude of these two-particle interactions determines the symmetry; a relatively modest Coulomb repulsion transforms an (anisotropic) *s*-wave symmetry into a *d*-wave symmetry [89]. These three features are in stark contrast with most alternative models for the description of the superconducting state in high temperature superconductors.

To be more specific, the BCS gap equation requires the knowledge of the electronic dispersion relation and the pairing potential. The in-plane parametrization of the dispersion for conduction holes, $\varepsilon_{\mathbf{k}}$, is well-established [56–58,102]. Neglecting here the interlayer hopping, the density of states $\rho_F = v_F^{-1}$ displays strong maxima along $\phi = n\pi/2$ ($n = 0, 1, 2, 3$) in the ($k_x; k_y$)-plane [56–58]. The strong anisotropy bears resemblance to the superconducting order parameter anisotropy.

The pairing interaction resulting from the long range coupling to $A_{1g}$, $B_{1g}$ and $B_{2g}$ optical phonon modes was derived in Ref. [56–58] and take the form

$$V_{A_{1g}}\left(\mathbf{q}_{\|}\right) \simeq \frac{U_{A_{1g}}}{\kappa^2 + q_x^2 + q_y^2}, \tag{12}$$

$$V_{B_{1g}}\left(\mathbf{q}_{\|}\right) = \frac{U_{B_{1g}}}{\kappa^2 + q_x^2 + q_y^2}\left(\cos k_x - \cos k_y\right)^2. \tag{13}$$

Here $\mathbf{q} = \mathbf{k} - \mathbf{k}'$ and $\mathbf{k} = (k_x, k_y)$ is the wave-vector component *parallel* to the CuO$_2$ plane. The coupling to the B$_{2g}$ mode is obtained from $V_{B_{1g}}$ rotating the $\mathbf{k}$-space basis by $\pi/4$. Because of this rotation the squared parenthesis in Equation (13) is small in the antinodal regions and vice versa. As a result, the contribution of this mode can be neglected. Hence, the maximum of the gap, $\Delta_{\max}$ and $T_c$ are determined by the A$_{1g}$ and B$_{1g}$ contributions to pairing. $U_{A_{1g},B_{1g}}$ and $\kappa$ were estimated in Ref. [89] to be $U_{A_{1g},B_{1g}} \sim 100$ meV and $\kappa \sim 0.3$ and are the same parameters that allow describing the experimentally observed phonon renormalization at the superconducting transition [103–105].

To solve the BCS gap equation one needs also to add the pairing interaction resulting from the short-range part of the electron-phonon interaction $H_{e-ph,S}$ [89]. Their expression is the standard dominant pairing interaction found in conventional superconductors with its attractive and repulsive parts.

Using both the long-range and short-range contributions of the electron-phonon interaction and Coulomb repulsion we solved the gap Equation [56–58,89]. Two important conclusions were obtained. First, the *magnitude* $\Delta_{\max}$ and *anisotropy* of the superconducting gap $\Delta_{\mathbf{k}}$ are determined by the coupling of charge carriers to the long range electron-phonon interactions at small $\mathbf{q}$. The anisotropic density of states indeed determines the $\mathbf{k}$-dependence of the gap. The results are in excellent agreement with the experimental determination of the order parameter. Second, the *symmetry* of the order parameter is not determined by the pairing interaction but by the relative weight of competing attractive and repulsive interactions. The long-range electron-phonon interactions lead to an anisotropic *s*-wave gap. Accounting for a relatively modest local Coulomb repulsion transforms the *s*-wave gap to a *d*-wave gap, without fundamentally affecting the magnitude or the overall $\mathbf{k}$-dependence of the gap [89].

### 4.2. Pair Fluctuations in the Symmetry-Broken Ground State

Motivated by the coexistence of stripe order and a two-dimensional *d*-wave like gap in the stripe phase of LBCO we proceed by investigating the structure of pairing fluctuations for static, metallic stripes, i.e., deep in the symmetry broken ground state. In this regard our present study is in some sense complementary to the work of Ref. [52] where SC has been obtained from ICDW fluctuations on top of a *homogeneous* ground state. The problem is complex because the formation of stripes also alters the spectrum of low energy charge and spin fluctuations which contribute to the correlations in the pairing channel. Due to this complexity we will use the time-dependent Gutzwiller approach, [106] instead of the SCDT, which conveniently allows also considering symmetry-broken solutions.

Our investigations are based on the one-band Hubbard model with hopping restricted to nearest ($\sim t$) and next nearest ($\sim t'$) neighbors

$$H = -t \sum_{\langle ij \rangle, \sigma} c_{i,\sigma}^\dagger c_{j,\sigma} - t' \sum_{\langle\langle ij \rangle\rangle, \sigma} c_{i,\sigma}^\dagger c_{j,\sigma} + U \sum_i n_{i,\uparrow} n_{i,\downarrow}. \tag{14}$$

Here $c_{i,\sigma}^{(\dagger)}$ destroys (creates) an electron with spin $\sigma$ at site $i$, and $n_{i,\sigma} = c_{i,\sigma}^\dagger c_{i,\sigma}$. $U$ is the on-site Hubbard repulsion.

As a starting point we treat the model Equation (14) within an unrestricted Gutzwiller approximation (GA) as in Ref. [106]. Basically one constructs a Gutzwiller wave function $|\Psi\rangle$ by applying a projector to a Slater determinant $|SD\rangle$ which reduces the double occupancy. The Slater determinant is allowed to have an inhomogeneous charge and spin distribution describing generalized spin and charge density waves determined variationally [107]. The advantage of the GA in the present context is that our saddle point solutions reproduce several features of experiments [106,108] while the same would not be true if the starting point where HF [106] for which stripes are not even the ground state for realistic parameters.

The parameters were fixed by requiring that (a) the linear concentration of added holes is $1/(2a)$ according to experiment [23,109,110] and (b) a TDGA computation of the undoped AF insulator reproduces

the experimental magnon dispersion [111] observed by inelastic neutron scattering. Condition (a) was shown to be very sensitive to $t'/t$ [106] whereas condition (b) is sensitive to $U/t$ and $t$, the former parameter determining the observed energy splitting between magnons at wave-vectors $(1/2, 0)$ and $(1/4, 1/4)$. Ref. [111] Indeed, the splitting vanishes within spin-wave theory applied to the Heisenberg model which corresponds to $U/t \to \infty$. We find that both conditions are met by $t'/t = -0.2$, $U/t = 8$ and $t = 354$ meV.

The results shown in this paper are for $d = 4$ bond-centered stripes oriented along the y-direction calculated on a $40 \times 40$ lattice; see Figure 5 for a visualization of the charge and spin structure. Figure 6 shows the corresponding band structure. Stripe formation induces two bands ($B_1$ and $B_2$, cf. Figure 6) in the Mott-Hubbard gap and the chemical potential is located in the (half-filled) band labeled $B_1$.

Dynamical pairing fluctuations are computed on top of the inhomogeneous solutions within the time-dependent GA [112,113] (TDGA). This scheme allows for the incorporation of particle-particle correlations in a similar manner as the traditional ladder approximation based on Hartree-Fock (HF) ground states. At the same time, it starts from a solution which incorporates correlations already at mean-field level. In the particle-hole channel the TDGA has previously been shown to provide an accurate description of magnetic fluctuations [114,115] and the optical conductivity [116] in cuprates.

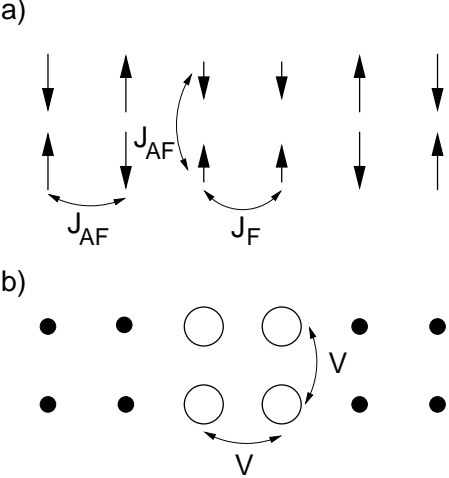

**Figure 5.** Spin (**a**) and charge (**b**) structure of $d = 4$ bond centered stripes at $\delta = 1/8$. Also shown are the interaction parameters used in the spin ($J_{AF}$, $J_F$) and charge ($V$) channel which determine the pairing fluctuations in Equation (16).

Here we are interested in the dynamical order parameter correlation function

$$P_q(\omega) = \frac{1}{\iota} \int_{-\infty}^{\infty} dt e^{\iota \omega t} \langle \mathcal{T} \Delta_q(t) \Delta_q^\dagger(0) \rangle, \tag{15}$$

where $\Delta_q = 1/\sqrt{N} \sum_k \gamma_k c_{-k-q,\downarrow} c_{k,\uparrow}$ and $\gamma_k$ specifies the symmetry of the order parameter fluctuations. We focus on $\gamma_k = (\cos(k_x) + \cos(k_y))/2$ (extended $s$-wave) and $\gamma_k = (\cos(k_x) - \cos(k_y))/2$ ($d$-wave) symmetries. For $\omega > 0$ ($\omega < 0$) the imaginary part of $P_q(\omega)$ yields the order parameter correlations for two-particle addition (removal). Please note that a pole in $P_q(\omega \to 0)$ signals the occurence of a long-range ordered SC state for a given symmetry. Of interest is also the vertex contribution $\Delta P_q(\omega) = P_q(\omega) - P_q^0(\omega)$ where $P^0$ corresponds to the non-interacting pair-correlation function calculated with the bare Green's function on the GA level. For a given symmetry $\Delta P_q(\omega)$ yields information on whether the order parameter fluctuations at a given momentum and frequency are attractive or repulsive for the GA quasiparticles.

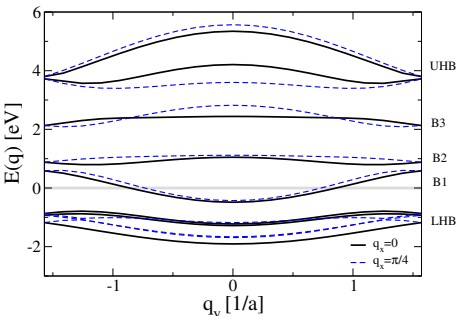

**Figure 6.** GA band structure for $d = 4$ bond-centered stripes oriented along the y-direction. Shown are the dispersions for $q = 0$ (black, solid) and $q = \pi/4$ (dashed). The energy is measured with respect to the chemical potential (horizontal grey line) and the lattice constant is set to $a \equiv 1$. LHB/UHB: Lower and upper Hubbard band.

Our ladder-type approach incorporates the pair correlations on a scale of $U$ but unfortunately does not properly take into account the energetically low lying collective excitations in the charge and spin particle-hole channels. As shown for the case of ICDW scattering [22] these excitations couple back to the particle-particle channel and thus can strongly enhance the low energy SC fluctuations. We incorporate this effect by adding the operator

$$
\begin{aligned}
H_{Fluc} &= -g \sum_{ij} J_{ij} \delta \langle c_{i,\uparrow}^\dagger c_{j,\downarrow}^\dagger \rangle \delta \langle c_{i,\downarrow} c_{j,\uparrow} \rangle \\
&+ g \sum_{ij} V_{ij} \delta \langle c_{i,\uparrow}^\dagger c_{j,\downarrow}^\dagger \rangle \delta \langle c_{j,\downarrow} c_{i,\uparrow} \rangle.
\end{aligned}
\tag{16}
$$

to the system which generates particle-particle scattering in the spin $((J_{ij}))$ and charge $((V_{ij}))$ channel but does not alter the ground state solution. The parameter $g$ is introduced to model vertex corrections which for simplicity are assumed to be constant. The interaction parameters $J_{ij}$ in the spin channel are obtained by calculating the magnetic excitations in the stripe phase within the TDGA [114,115]. The low energy Goldstone mode emerging from the incommensurate wave-vectors can be fitted by linear spin-wave theory applied to a Heisenberg model with site dependent interactions as shown in Figure 5a. We find $J_{AF} = 0.4t$ between antiferromagnetically ordered spins and $J_F = -0.2J_{AF}$ between the ferromagnetically ordered bonds on the stripe legs. To obtain information about the interaction parameters in the charge channel we calculate the charge profile of bond centered stripes within the following HF approximated spinless fermion model

$$
H = -t \sum_{\langle ij \rangle} f_i^\dagger f_j - t' \sum_{\langle\langle ij \rangle\rangle} f_i^\dagger f_j + \sum_{ij} V_{ij}(1 - n_i)(1 - n_j).
$$

where the kinetic part is the same as in Equation (14) and $V_{ij}$ is a nearest-neighbor attraction acting on holes on the stripe legs (Figure 5b). We find that a value of $V_{ij} = -0.25t$ reproduces the charge profile obtained within the full GA calculation.

Figure 7a shows the instantaneous pairing correlations in the $d$-wave channel obtained from the removal part $\langle \Delta_q^{\dagger,d} \Delta_q^d \rangle = \int_{-\infty}^0 d\omega P_q(\omega)$ for coupling parameter $g = 1$. The correlations calculated from the addition spectra $(\omega > 0)$ are similar. The shape reflects the quasi one-dimensionality of the underlying ground state with the most pronounced correlations along $(q_x, q_y \approx 0)$ and the maximum at $(0, 0)$. However, when we substract the contribution of the non-interacting GA quasiparticles the resulting vertex contribution (cf. Figure 7b) takes a different and much more isotropic shape. The correlations at $\mathbf{q} = (0, 0)$

are still attractive but the maxima now occur at $(\pm\pi, 0)$ and $(0, \pm\pi)$. The vertex contribution for extended *s*-wave symmetry (cf. Figure 7c) is also rather isotropic but displays the maximum attraction at $\mathbf{q} = (0,0)$. Also the (instantaneous) attraction is by a factor of 3–4 larger than in the *d*-wave channel. The strong reduction of the *d*-wave attractive fluctuations for small momenta can be traced back to the form factor $\gamma_k = \cos(k_x) - \cos(k_y)$ and occurs also in the homogeneous state in the absence of $J_{AF}$, $J_F$, and $V$ (for finite interaction parameters the homogeneous state has a pole at $\mathbf{q} = 0$ and $\omega = 0$ reflecting an instability towards superconductivity).

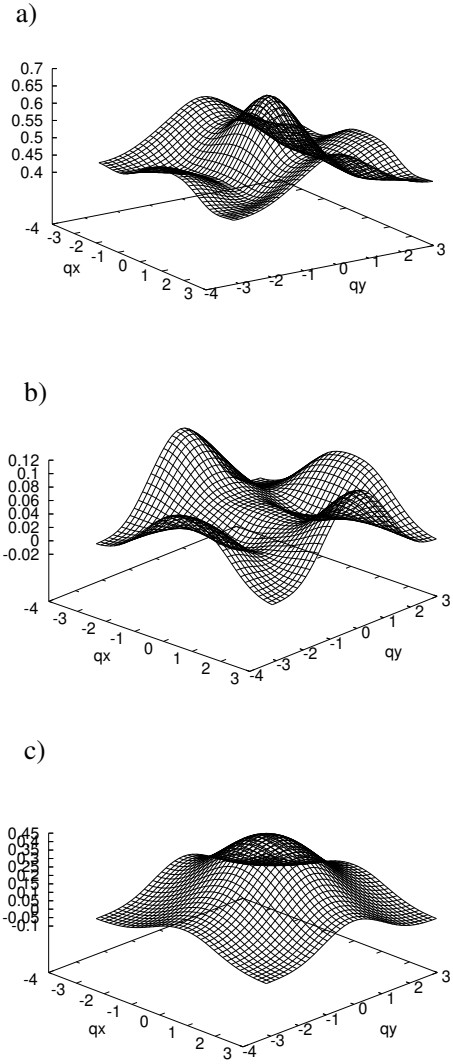

**Figure 7.** Instantaneous order parameter fluctuations for *d*-wave (**a**,**b**) and extended *s*-wave symmetry (**c**) obtained from the removal part ($\omega < 0$). Panel (**a**) corresponds to the TDGA ladder result for $\langle \Delta_q^d \Delta_q^d \rangle = \int d\omega P_q(\omega)$ whereas panels (**b**) and (**c**) show the vertex contributions $\int d\omega \Delta P_q(\omega)$. Coupling parameter $g = 1$.

It is important to know in which frequency range $P_q(\omega)$ contributes most to the instantantaneous correlations. One can expect that a system with low energy order parameter fluctuations is more susceptible towards the transition to a superconducting state than if these would occur at higher frequencies. In fact,

we find pronounced differences between *d*-wave and extended *s*-wave symmetry as can be seen from Figure 8. In the main panels we show the $\mathbf{q} = 0$ pairing fluctuations for two-particle removal ($\omega < 0$) and addition ($\omega > 0$) and a coupling parameter ($g = 1$). Also shown are the bare GA two-particle spectra which correspond to a convolution of the single-particle bands shown in Figure 6. In the *d*-wave sector (panel a) one observes two features in the uncorrelated removal part. The lower (higher) energy one is due to the removal of two particles from the band $B_1$ ($B_2$) whereas interband correlations (which have weight in between) are suppressed. For extended *s*-wave fluctuations (panel b) the GA removal part has only weight in the energy range which corresponds to the removal of two particles from $B_2$.

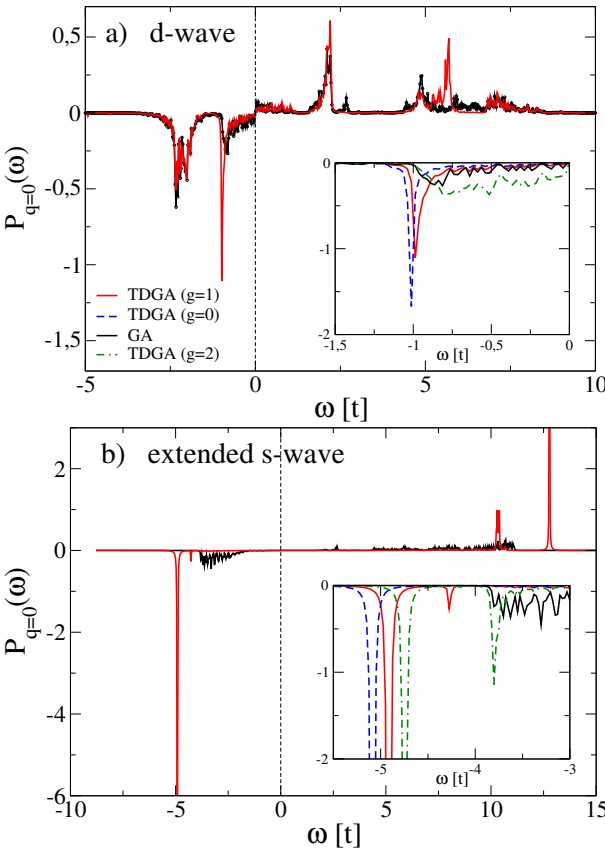

**Figure 8.** Order parameter fluctuations at $\mathbf{q} = 0$ for *d*-wave (**a**) and extended *s*-wave symmetry (**b**). Red: TDGA; Black: GA. The inset covers the energy range around the bound states of the removal spectra. Also shown (blue dashed) is the TDGA result without spin and charge attractive interactions ($g = 0$) and the strong coupling case (green dashed dotted) for $g = 2$.

Including correlations within the TDGA ladder scheme leads to the appearance of bound ($\omega < 0$) and antibound ($\omega > 0$) states similar to the physics of local pairs in the Hubbard model as relevant for e.g., Auger spectroscopy [112,117,118]. In case of the *d*-wave removal fluctuations the bound state forms at the bottom of the convoluted band $B_1$. As can be seen from the inset to Figure 8a this bound state exists even without the inclusion of additional interactions in the spin and charge channel ($g = 0$). The effect of $J_F$, $J_{AF}$ and $V$ is to push the bound state toward lower energy, thus enhancing the mixing with the $B_1$ band states and strengthening the *d*-wave correlations inside the active band. In fact, for $g = 2$ the bound state is no more visible as a separate feature and instead one observes a convoluted $B_1$ band with increased *d*-wave correlations as compared to the $g = 0$ case.

By contrast, the bound state in the extended *s*-wave removal part occurs below the convoluted lower Hubbard band (LHB, cf. Figure 6). Also in this case inclusion of $J_F$, $J_{AF}$ and $V$ leads to a shift of the bound state to lower energies and induces also a smaller satellite. Upon increasing the coupling ($g = 2$) this satellite increases in intensity and approaches the energy of the LHB. However, the formation of the extended *s*-wave bound state is accompanied by a strong suppression of two-hole band states in contrast to the *d*-wave case so that there are no low energy band states with extended *s*-wave correlations.

## 5. Conclusions

In this work, we investigated the mechanism of phase separation in systems with strong electron correlations. The perturbation series expansion around the atomic limit is a reasonable approach for the description of such systems. Therefore, processes of atomic-level crossing occurring at certain values of the chemical potential play a key role in the evolution of their band structure with doping. An extreme non-rigidity of the bands near these peculiar chemical potentials leads to the appearance of regions of negative electron compressibility. Hence these regions are an inherent property of all strongly correlated systems. We have demonstrated their occurrence in the one-band Hubbard and Hubbard-Kanamori models widely used for the description of cuprate perovskites, transition metal oxides, and some other crystals. The existence of these regions gives rise to the charge instability—the segregation of the crystal into electron-rich and electron-poor domains. The energy released by their formation has to be absorbed by phonons. This leads to different lattice distortions in the above domains. In the case of a predominant interaction of electrons with distortions of special symmetry, for example, with a softening phonon mode, the separation into two domains acquires the shape of stripes.

For such textures we have investigated the frequency and momentum structure of pairing correlations in the *d*-wave and extended *s*-wave channels. It turns out that depsite the underlying quasi one-dimensional electronic structure these correlations are quite isotropic, similar to the isotropy of spin fluctuations which arise from stripe textures, [114,115] which therefore is compatible with the observation of an isotropic superconducting gap in stripe ordered compounds [35]. While the present approach was restricted to the evaluation of pairing fluctuations on top of the normal state it would be interesting in future work to investigate directly the structure of superconducting order in the symmetry broken stripe state.

**Author Contributions:** Conceptualization, R.K.K. and G.S.; formal analysis, A.B., V.H., G.S., A.S. (Aleksander Shelkan) and A.S. (Alexei Sherman); software, A.S. (Aleksander Shelkan); writing—original draft, G.S.; writing—review & editing, A.B., V.H., R.K.K., G.S. and A.S. (Alexei Sherman). All authors have read and agreed to the published version of the manuscript.

**Funding:** V.H.'s contribution was supported by the grant of the Estonian Scientific Council PRG347. G.S. acknowledges support from the Deutsche Forschungsgemeinschaft.

**Conflicts of Interest:** The authors declare no conflict of interest.

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
