# Peer review of "Phase Separation and Pairing Fluctuations in Oxide Materials"

_condensedmatter, doi:10.3390/condmat5040065_

Round 1
Reviewer 1 Report
The article summarizes the results of many years of research of the phenomenon of phase separation in layered cuprates by a large group of authors. What is new is the study of the role of order parameter fluctuations. In the framework of the model, taking into account strong electronic correlations, it is demonstrated that the coexistence of stripe fluctuations and superconductivity of d-type is possible.Author Response
We appreciate the positive comment by the
referee on our work.
Reviewer 2 Report
The article by Andreas Bill et al., titled "Phase separation and pairing fluctuations in oxide materials", investigates at a theoretical and computational level the mechanism of phase separation of charges in physical systems with strong electronic correlations.
In this article, specific calculations have been done for two different hamiltonians: (i) the one-band Hubbard hamiltonian widely used for the description of cuprate perovskite / superconductors and (ii) the
Hubbard-Kanamori hamiltonian, which with the inclusion of the Hund term to describe the role of the spin and orbital degrees of freedom, is more suitable to be applied to transition metal oxides and iron-based superconductors, characterized by diverse orbitals contributing to the electronic properties.
One of the main outcome of the calculations in this article is the finding of regions of negative electron compressibility, and hence charge instability with segregation in regions of electron-rich and electron-poor domains at peculiar values of the chemical potential.
The Authors then investigate the interesting case of electrons interacting in a predominant manner with lattice distortions of special symmetry. In this relevant case the phase separation into two domains of charges acquires the geometrical shape of stripes.
This is the most interesting and key result reported in the article, in connection with the phenomenology of superconducting cuprates:
by investigating the frequency and momentum dependence of pairing
in the d-wave and in the s-wave channels in the normal state,
depsite the underlying stripe electronic structure, the pairing is isotropic,
compatible with the observation of an isotropic superconducting gap in stripe ordered compounds.
I found this result of great importance from a conceptual point of view and for the descrition of the physics of superconducting cuprates and related compounds.
I would like to suggest to the Authors to include a few references from the Bianconi's group. Indeed, Bianconi et al. have proposed since the beginning of the cuprate era the key role of static stripes in modifing the electronic structure, inducing a deep reconfiguration of the pairing, and in enhancing the critical temperature of the superconducting state.
E.g.:
The gap amplification at a shape resonance in a superlattice of quantum stripes: A mechanism for high Tc
A Perali, A Bianconi, A Lanzara, NL Saini
Solid State Communications 100, 181-186, 1996
High Tc superconductivity in a superlattice of quantum stripes
A Bianconi, A Valletta, A Perali, NL Saini
Solid State Communications 102, 369-374, 1997
Anomalous isotope effect near a 2.5 Lifshitz transition in a multi-band multi-condensate superconductor made of a superlattice of stripes
A Perali, D Innocenti, A Valletta, A Bianconi
Superconductor Science and Technology 25, 124002, 2012
Once the Authors have considered the above discussed expansion of the bibliography, the article by Andreas Bill eta can be accepted for publication in Condensed Matter.
Author Response
We would like to thank the referee for her/his
positive evaluation of our work and for pointing
out the relevance of some references which we missed
in the previous version of our manuscript.
We have followed the referees suggestion and incorporated
these references from the Bianconi group.
Reviewer 3 Report
The paper "Phase separation and pairing fluctuations in oxide materials" by Andreas Bill, Vladimir Hizhnyakov, Reinhard Kremer et al is close to the scientific interests of Prof. Alex Muller and even contains 2
references on the joint papers of Prof. Muller with the group of one of
the coauthors of this paper Dr. Hizhnyakov.
The paper deals with the actual topics in physics of strongly correlated electron systems namely of phase-separation , formation
of stripes and pairing fluctuations in cuprates and related
materials.
First of all the authors are considering the important
question how quasi-one dimensional stripes can promote quasi-two
dimensional superconductivity in cuprates.
From theoretical side the authors utilize mean-field type (HF)
approaches and diagrammatic methods of power series expansion around
the atomic limit for the Hubbard model and its multiband generalizations
such as Hubbard-Kanamori models.
The regions of phase-separation are defined as usual by the negative
electron compressibility and Maxwell-construction
for the chemical potential.
We can discuss the reliability of the mean-field approaches and
convergence of the series expansions around the atomic limit at strong
coupling ( large U) and close to half-filling ( n=1-delta) for the
Hubbard model but to some extent we do not have many options here since
in the difficult regions of the phase diagram typical for cuptates exact
diagrammatic methods are often not applicable or can be used only as an
extrapolation.
Still the authors should compare their results on phase-separation and pairing fluctuations here with different approaches specified e.g. by slave-boson or slave-fermion approximations or large-N expansions.
Allso a set of important
references on d-wave pairing and phase-separation in manganites and
cuprates is missing and should be added to the Reference list and
briefly discussed in the article.
As a minimal set I strongly recommend the following articles:
1) M.Yu. Kagan and T.M. Rice, "Superconductivity in the two-dimensional
t-J model at low electron density", Jour.Phys.:Condens.Matter, v.6,
p.2471, 1994
2) M.Yu.Kagan, K.I. Kugel, " Inhomogeneous charge distributions and
phase-separation in manganites", Physics Uspekhi, v.44, p.553, 2001
3) K.I. Kugel, D.I. Khomskii, " The Jahn-Teller effect and magnetism:
Transition metal compounds" , Sov.Phys. Uspekhi, v.25, p.231, 1982
4) A.O. Sboychakov, K.I. Kugel and A.L. Rakhmanov, " Phase-separation in
a two-band model for strongly correlated electrons", Phys.Rev.B,v.76,
p.195113, 2007 , and related papers of the authors where they considered phase-separation in the model in Hubbard-I approximation.
My conclusion the article can be published with minor revision. The authors should take into account the referee remarks.
Author Response
We would like to thank the referee for her/his
positive evaluation of our work and her/his
helpful comments which we have considered in our
revised manuscript.
The referee suggests to relate our investigations
on phase separation to those of previous works and also
for other systems. We
have therefore added corresponding references (cf. lines
140-143) in the revised manuscript, among them also those
proposed by the referee.
In response to the referees comment:
"We can discuss the reliability of the mean-field approaches and
convergence of the series expansions around the atomic limit at strong
coupling ( large U) and close to half-filling ( n=1-delta) for the
Hubbard model but to some extent we do not have many options here since
in the difficult regions of the phase diagram typical for cuptates exact
diagrammatic methods are often not applicable or can be used only as an
extrapolation."
we have added a remark in lines 152 - 155 of the revised manuscript.
Round 2
Reviewer 2 Report
The revised version of the article by Andreas Bill et al. results improved in terms of presentation and discussion of the relevant literature, also in an historical perspective. The Authors have taken in due consideration all the suggestions reported during the review process.
Therefore, I can recommend for publication in Condensed Matter the article by Andreas Bill et al. in the present form.
Reviewer 3 Report
The article is significantly improved after the revisions. All the remarks of the referee are taken into account . I have no further remarks andcan recommend the article in the present form for the publication in the journal Condens.Natter,